# Silica deposits on Mars with features resembling hot spring biosignatures at El Tatio in Chile

Steven W. Ruff[1] & Jack D. Farmer[1]

The Mars rover Spirit encountered outcrops and regolith composed of opaline silica (amorphous $SiO_2 \bullet nH_2O$) in an ancient volcanic hydrothermal setting in Gusev crater. An origin via either fumarole-related acid-sulfate leaching or precipitation from hot spring fluids was suggested previously. However, the potential significance of the characteristic nodular and mm-scale digitate opaline silica structures was not recognized. Here we report remarkably similar features within active hot spring/geyser discharge channels at El Tatio in northern Chile, where halite-encrusted silica yields infrared spectra that are the best match yet to spectra from Spirit. Furthermore, we show that the nodular and digitate silica structures at El Tatio that most closely resemble those on Mars include complex sedimentary structures produced by a combination of biotic and abiotic processes. Although fully abiotic processes are not ruled out for the Martian silica structures, they satisfy an *a priori* definition of potential biosignatures.

[1] School of Earth and Space Exploration, Arizona State University, Tempe, Arizona 85287-6305, USA. Correspondence and requests for materials should be addressed to S.W.R. (email: steve.ruff@asu.edu).

Hydrothermal spring deposits of silica (sinter) have long been targets in the search for fossil life on Mars[1] and early Earth[2] because of their ability to capture and preserve biosignatures. Exposures of opaline silica were first discovered on Mars in 2007 by the Spirit rover adjacent to the 'Home Plate' feature in the inner basin of the Columbia Hills of Gusev crater[3]. The silica occurs most commonly in nodular masses that have a rubbly appearance but are considered outcrops because of their stratiform expression and resistance to deformation by the rover wheels[4] (Fig. 1). These outcrops are found in a patchy distribution, commonly overlying a platy bedrock unit dubbed Halley Subclass that has been interpreted as an altered ash deposit[3]. Volcanic lapillistone also is seen to overlie this unit and is capped by vesicular basalt. The presence of opaline silica in the context of a succession of basaltic volcanic rocks has been interpreted as evidence of past volcanic hydrothermal activity[3].

On Earth, fumaroles are one manifestation of hydrothermal activity in addition to hot springs and geysers. Acid-sulfate steam condensates produced by fumaroles have the capacity to leach metal cations from basaltic rocks, leaving behind a residue of opaline silica. This process was hypothesized for the origin of Home Plate silica in the work of Squyres *et al.*[3], and favoured over the alternative hypothesis of silica precipitation from neutral-to-alkaline hot spring fluids. A Ti enrichment in the silica-rich regolith occurrence was viewed as supporting evidence for acid leaching because Ti is relatively immobile under such conditions. The nodular expression of the silica outcrops was noted but not interpreted in that work.

A subsequent study by Ruff *et al.*[4] presented observations of the silica outcrops that support a hot spring and/or geyser origin, including: their typical overlying stratiform relationship with a local rock unit (Halley Subclass) with no apparent crosscutting or fracture controlled occurrences; and their unique morphology and textures that cannot be tied to any of the potential precursor lithotypes in the exposed stratigraphic section. Additionally, it was noted that the Ti content of silica-rich materials does not uniquely constrain their origin to acid-sulfate leaching given that relatively Ti-rich silica sinters are known to occur on Earth[5]. The nodular and digitate structures of the Home Plate silica outcrops were recognized as common features and tentatively interpreted as the result of aeolian erosion[4].

Thermal infrared emission spectra of the Home Plate silica outcrops obtained by Spirit's Miniature Thermal Emission Spectrometer (Mini-TES; $\sim 340$–$2,000\, cm^{-1}$) were used to identify the opaline silica (opal-A) component[3] and distinguish it from other silica polymorphs[4]. However, these spectra commonly display a strong absorption feature near $1,260\, cm^{-1}$ that typically is weak or absent in terrestrial opaline silica[3,4]. Previous work demonstrated that this feature varies as a function of viewing geometry[6] and that opaline silica measured at high emission angles ($>45°$) results in a feature that in some cases has a depth sufficient to match that in some of the Mini-TES spectra[3,4].

Although the mast-mounted configuration of Mini-TES led to an optical beam path that intersected horizontal surfaces at angles $>40°$ from normal[7], in no case were the nodular silica outcrops smooth and flat lying, especially across the $>10\, cm$ diameter field of view of Mini-TES. Thus the potential for achieving high emission angle viewing geometry in these observations was unlikely, making this explanation for the unusual opaline silica spectra less certain.

New observations of silica sinter deposits from the active volcanic hydrothermal system at El Tatio provide a basis for scale-integrated comparisons to the previously identified silica features at Home Plate, including geologic context, mesoscale structures in outcrops, mm-scale textures, and spectral signatures.

The physical environment of El Tatio offers a rare combination of high elevation ($\sim 4,300\, m$), low precipitation rate ($<100\, mm$ per year), high mean annual evaporation rate ($132\, mm$), common diurnal freeze-thaw[8] and extremely high UV irradiance[9]. Such conditions provide a better environmental analog for Mars than those of Yellowstone National Park (USA) and other well-known geothermal sites on Earth. Our results demonstrate that the more Mars-like conditions of El Tatio produce unique deposits, including biomediated silica structures, with characteristics that compare favourably with the Home Plate silica outcrops. The similarities raise the possibility that the Martian silica structures formed in a comparable manner.

## Results

**Field observations.** Hot spring and geyser discharge channels at El Tatio commonly contain nodular masses of opaline silica sinter (Fig. 2). Many of these silica nodules display mm-scale digitate structures that are strikingly similar in overall form to those adjacent to Home Plate (Fig. 3). Given the volcanic hydrothermal setting and presence of opaline silica at both sites, the qualitative similarities in size and shape of the silica nodules and their digitate structures leads to the hypothesis that they may have formed through similar processes. Many El Tatio nodules are silica coated and cemented breccias composed of reworked pebbles of older, locally derived volcanic rocks and fragments of silica sinter. Breccia clasts become coated by laminated opaline silica via silica precipitation during transport along outflow channels, and are then subject to further fragmentation during cycles of transport and cementation. This produces pebble to cobble-sized breccias containing a complex association of volcaniclastic and silica sinter materials with diverse internal textures and compositions that reflect local sediment sources. Breccias that line channel floors and margins provide the substrate upon which digitate structures form (Supplementary Fig. 1).

El Tatio discharge channels that host nodular, digitate sinter typically have shallow ($<5\, cm$ depth) flowing water that supports microbial biofilms and mats containing a diverse assemblage of diatoms and filamentous cyanobacteria, where water temperature is $<40°C$ (ref. 10). Water pH is circum-neutral ($\sim 6.5$–$7.5$) throughout El Tatio[11]. The aspect ratio, shape, and spatial density of the nodular and digitate structures vary among different channels and even along flow paths within a given channel (Supplementary Fig. 2), likely due to differences in depth, flow direction and velocity and the microenvironmental conditions created by microbial communities[12]. Morphologic variations also are evident among the Home Plate silica structures (Figs 1 and 3a,c,e), perhaps indicative of similar variations in depositional conditions.

Some of the mm-scale textural features of El Tatio silica sinters appear to have counterparts among Home Plate nodular silica outcrops. One of the Home Plate outcrops was intentionally disturbed with the rover's wheel (Fig. 1d), producing broken and/or overturned fragments with exposed interior and/or underside surfaces that likely are relatively pristine. These fragments were investigated using the Microscopic Imager (MI), a camera mounted on the rover's arm capable of grayscale imaging at $\sim 30\, \mu m$ per pixel (ref. 13). As shown in previous work, one of the fragments, dubbed Norma Luker, displays a texture suggestive of a sinter breccia[4], which is common among terrestrial sinter deposits including those at El Tatio (Fig. 4a,b). A second fragment, dubbed Innocent Bystander, displays a pervasive microporous surface texture with coated grains[4] that together resemble a sample of El Tatio nodular silica sinter collected from a discharge channel (Fig. 4c,d). At El Tatio, this precipitated texture is less common than breccia textures and

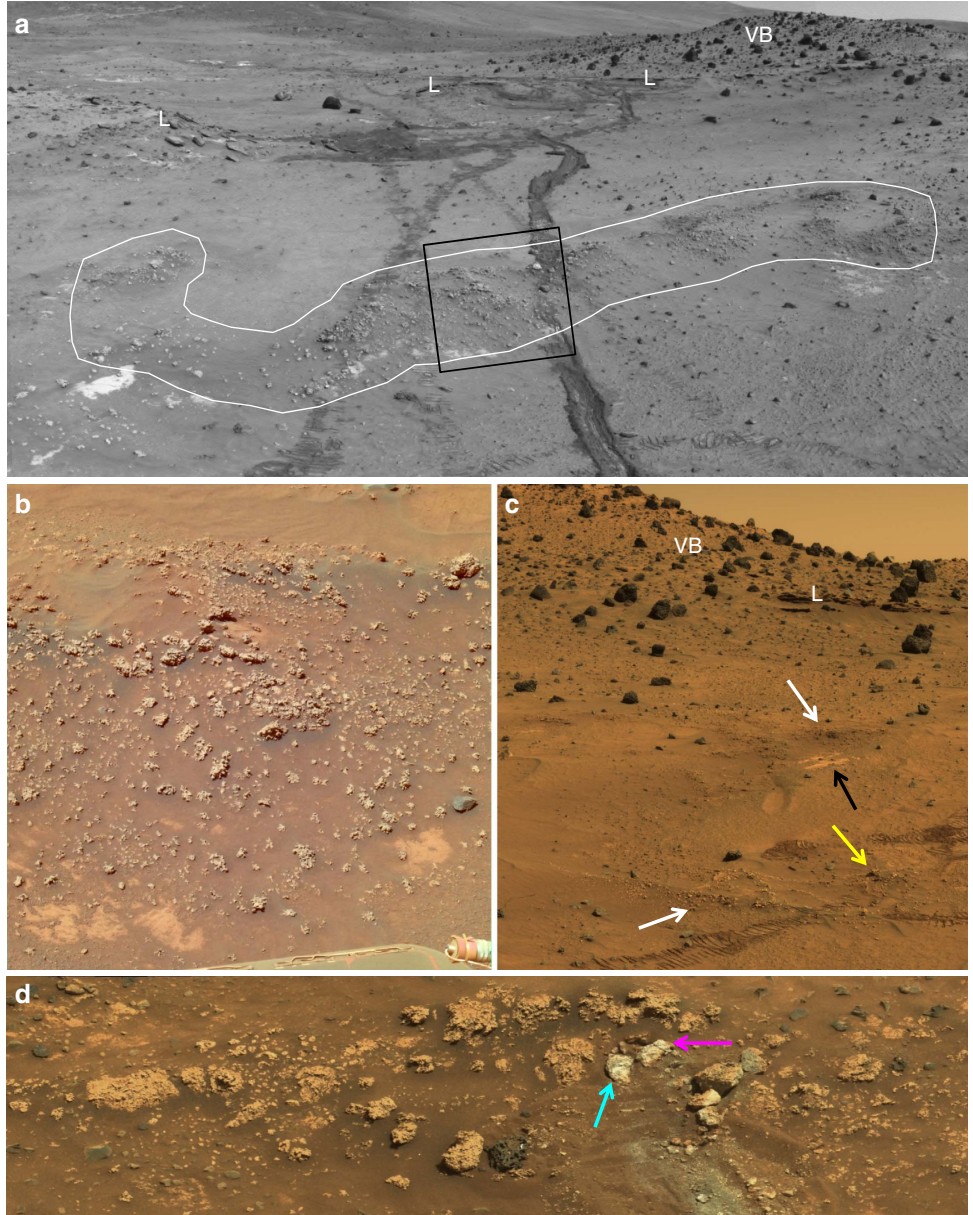

**Figure 1 | Opaline silica nodular outcrops adjacent to Home Plate showing typical stratiform expression.** (**a**) White outline highlights nodular silica outcrop (Navcam mosaic, sol 1116). Rover wheel tracks are ∼1 m apart. Rolling wheels did not deform the ∼15 cm high outcrop (lighter tracks) compared with the inoperative dragging wheel in a later traverse (darker track). Box indicates approximate location of **b**; L is lapillistone, VB is vesicular basalt. (**b**) Pancam approximate true colour (ATC; sol 778, P2388) of opaline silica nodules from **a**, before rover traverse. Midfield scene spans ∼80 cm. (**c**) Westerly view toward Low Ridge showing nodular outcrops (white arrows) over light-toned platy outcrop (black arrow) and location where rover wheels passed over nodular outcrop without disturbing it (yellow arrow); L and VB as in **a**. Midfield scene spans ∼3 m (cropped Pancam ATC mosaic, sol 800, P2401). (**d**) Nodular outcrop with portion intentionally disturbed by rover wheel (whitish hues). Cyan arrow indicates 'Innocent Bystander' and magenta arrow is 'Norma Luker', the two pieces investigated with the rover arm instruments and shown in Fig. 4. Scene spans ∼110 cm (cropped Pancam ATC mosaic, sol 1234, P2378).

appears limited to the underside of sinter nodules there. This may be consistent with Innocent Bystander representing the now exposed underside of an overturned fragment of the outcrop.

A porous, sponge-like texture was described previously for the Home Plate nodular silica outcrop dubbed Elizabeth Mahon, along with its smoother digitate protrusions[4] (Fig. 4e). The smoother portions were suggested to be the result of aeolian abrasion, but the porous texture was not interpreted. Close inspection raises the possibility that the appearance of porosity may be due in part to fine basaltic sand, evident elsewhere in the scene, trapped within roughness elements on the surface of the outcrop. We now recognize a candidate for the variably textured surfaces of Elizabeth Mahon among samples of El Tatio sinter, in which sub-mm roughness creates an irregular pattern that mimics the appearance of mm-scale porosity (Fig. 4e,f). The protruding features of the El Tatio sample are smoother, akin to those on Elizabeth Mahon. The variable texture of the El Tatio sample is actually a surficial fabric rather than a manifestation of porosity, as readily demonstrated by comparing this topside surface to the underside of the same sample, which displays unambiguous microporosity (Fig. 4d). Apparently silica accumulated on the top surface of the El Tatio sample via

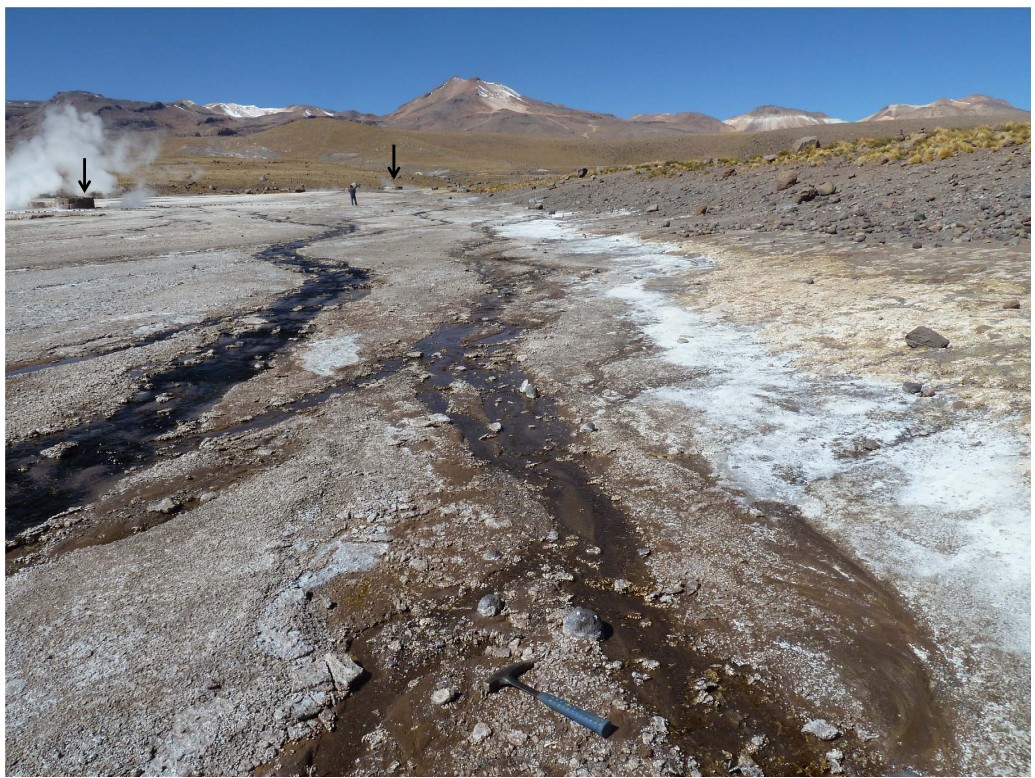

**Figure 2 | A portion of the volcanic hydrothermal system at El Tatio in Chile.** Discharge channels emanating from small ($\sim$1–3 m) steaming hot springs behind stone barricades (arrows) deposit silica sinter and efflorescent salts amidst volcanic detritus. Rock hammer is 33 cm long.

evaporative precipitation in a manner that obscures the bulk porosity and created variable roughness. Although this candidate textural analog does not preclude the possibility that aeolian abrasion is responsible for the variable texture of Elizabeth Mahon, it demonstrates that silica deposition alone can lead to a similar texture.

It is important to recognize that independent of whether we have identified the correct textural analog for this or other Home Plate silica outcrops, the presence of mm-scale textural variations seen among them is a characteristic consistent with what is seen in terrestrial silica sinter deposits. The varied textures of terrestrial sinters reflect the diverse depositional environments of hot spring/geyser systems over a range of spatial scales[14–16]. This also is true of the microscale internal textures of silica sinters ($<$100 μm), including El Tatio samples for which we present scanning electron microscopy (SEM) and petrographic thin section views in a subsequent section. Unfortunately, the resolution of Spirit's microscopic imaging capability precludes our ability to observe any microscale features among the Home Plate silica structures.

**Spectroscopy**. We have found that laboratory thermal infrared emission spectra of some silica sinter samples from El Tatio have a strong $\sim$1,260 cm$^{-1}$ feature independent of emission angle. The presence of this feature in some Mini-TES spectra of Home Plate silica was assumed to result from high emission angle viewing geometry[3,4] (Supplementary Fig. 3), but some El Tatio samples produce this feature at 0° emission angle, providing a good fit to Mini-TES spectra of some Home Plate silica outcrops (Fig. 5a). We attribute this spectral behavior to a thin (tens of micrometers) patchy crust of halite (NaCl) that coats sinter surfaces (Fig. 5b). The spectral contribution of halite is apparent by measuring the same sample before and after gentle scrubbing with a toothbrush

and deionized water. This action effectively removed halite from the surface without disturbing the silica, which was confirmed by SEM (Fig. 5c), elemental analysis using energy dispersive spectroscopy (EDS; Supplementary Table 1), and taste. Samples measured after halite removal display a feature shifted to $\sim$1,250 cm$^{-1}$ and substantially reduced in contrast, resulting in spectra notably similar to halite-free silica sinter, for example, from a hot spring in Yellowstone National Park (Fig. 5a).

Some Mini-TES spectra of Home Plate silica outcrops display a $\sim$1,260 cm$^{-1}$ feature with a depth and position not achievable from the viewing geometry effect alone but evident among halite encrusted El Tatio sinter samples independent of emission angle (Supplementary Fig. 4). Given that halite has no absorption features in this spectral range[17], the appearance of a strong $\sim$1,260 cm$^{-1}$ feature in halite-encrusted sinter samples is enigmatic and apparently has not been documented previously. Halite has an index of refraction of $\sim$1.5 near 1,260 cm$^{-1}$ (ref. 18) versus $\sim$0.5 for amorphous silica, which perhaps accentuates the known geometric effect.

The $\sim$1,260 cm$^{-1}$ feature observed in Mini-TES spectra of Home Plate silica outcrops ranges from strong to absent[4]. The presence or absence of a thin, patchy halite crust akin to that of El Tatio sinter could explain this variability. The detectability of the Na and Cl in such a crust by Spirit's Alpha Particle X-ray Spectrometer (APXS) is unknown, but would be dependent on its thickness and coverage. Unfortunately, none of the outcrop targets displaying a strong $\sim$1,260 cm$^{-1}$ feature was measured by the APXS, precluding a direct comparison between the two instruments.

**Microscopy**. Our investigation of the nodular and digitate silica structures from El Tatio using high vacuum and environmental scanning electron microscopy (SEM/ESEM) revealed internal microlaminations with fenestral porosity and silica encrusted

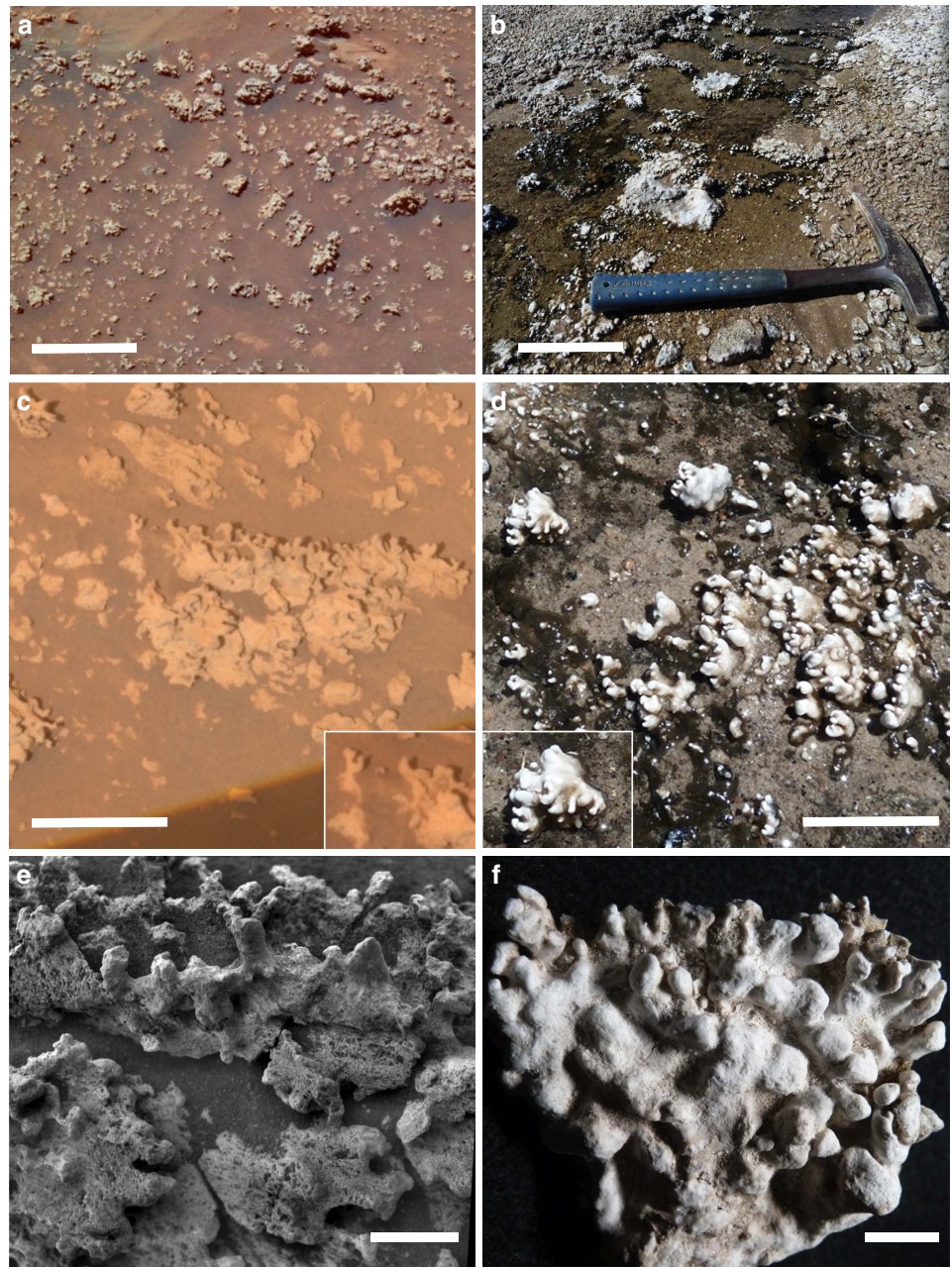

**Figure 3 | Comparison of opaline silica structures adjacent to Home Plate with those of hot spring discharge channels at El Tatio.** (**a**) Home Plate opaline silica occurs in nodular masses with digitate structures that resemble those at El Tatio (**b**), at the same scale. Mars scene is cropped from Fig. 1b. White scale bar in **a** and **b** represents 10 cm. (**c**) Home Plate opaline silica digitate structures resemble those at El Tatio (**d**) at the same scale. The white scale bar in **c,d** represents 5 cm. Insets highlight notably similar structures. Mars scene is a Pancam ATC image ('Elizabeth Mahon', sol 1160, P2582). Reddish hues in Mars scenes are due to thin airfall dust accumulation. (**e**) Grayscale Microscopic Imager mosaic (sol 1157) of a portion of the 'Elizabeth Mahon' silica outcrop on Mars shown in c has similar structures as those on sample ET1-1A from a hot spring discharge channel at El Tatio (**f**). The white scale bar in **e,f** represents 1 cm.

microbial biofilms with filaments, sheaths, and exopolymeric substances (EPS) on both internal and external surfaces (Fig. 6a,b). EDS showed C enrichment consistent with the presence of organic matter (Supplementary Table 1). In cross section, laminae alternate between non-porous silica, filamentous sinter, and open fenestrae comparable to microstromatolitic sinter from Yellowstone[19], New Zealand[20,21] and Iceland[22]. The role of microbial biofilms and their EPS in contributing to these microtextural features was demonstrated previously for some New Zealand siliceous microstromatolites[23]. Among El Tatio digitate silica structures, we have documented at least one

example where unsilicified EPS film is present in an especially large ($\sim 100 \times 1,000 \, \mu m$) fenestra (Supplementary Fig. 5).

Petrographic thin sections of El Tatio silica structures reveal textural and compositional complexity, reflecting a range of microenvironmental conditions during their formation. Finely laminated internal textures are evident, including both flat laminated and columnar forms of stromatolitic opaline silica (Fig. 6c) sometimes containing coated grains and pisoliths formed where silica laminae accreted onto angular clasts of porphyritic volcanics during transport (Supplementary Fig. 1). Fine laminae of clear opaline silica cement (tens of micrometers thick) typically

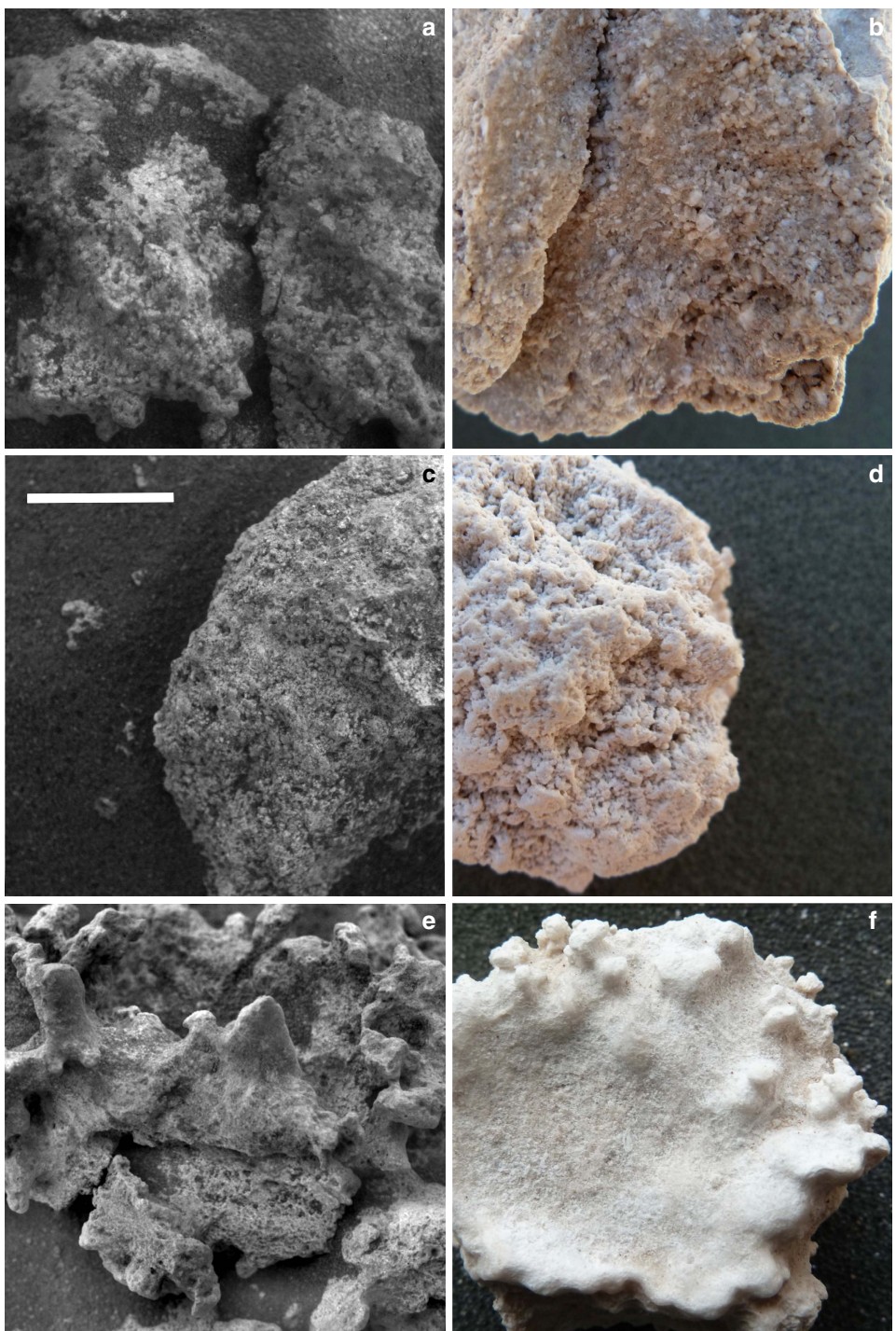

**Figure 4 | Comparison of mm-scale textural features of home plate opaline silica rocks and El Tatio silica sinter samples.** (**a**) Fragments dubbed Norma Luker (MI image, sol 1291) from the disturbed outcrop seen in Fig. 1d display a texture like that of El Tatio sinter breccia in **b**. (**c**) Another disturbed outcrop fragment, dubbed Innocent Bystander (MI image, sol 1251), displays microporosity and possible coated grains like that of the underside of El Tatio silica nodule in **d**. (**e**) The variably textured surfaces of Elizabeth Mahon (MI image, cropped from Fig. 3e) are similar to those of El Tatio silica nodule in **f**, which is the topside of the one shown in **d**. White scale bar represents 1 cm and applies to all images.

lack identifiable microfossils. However, they alternate with thicker laminae displaying fenestral cavities that contain fine, silica encrusted filamentous microfossils and empty sheaths. Finally, columnar forms include discrete laminae that contain a wide variety of unidentified filamentous and coccoidal biomorphs, diatom frustrules, and occasionally, local populations of heavily ensheathed fossil cyanobacteria (Fig. 6d) resembling *Calothrix* (family Rivulariaceae)[24].

Thin sections of El Tatio sinters commonly display laterally persistent, lenticular to wavy laminae dominated by distinctive palisade microtextures oriented roughly perpendicular to laminae (Fig. 6c,d). The palisades are dominated by heavily ensheathed, *Calothrix*-like filamentous cyanobacteria that sometimes alternate with thinly laminated intervals containing finely filamentous microfossils with recumbent orientations, parallel to laminations. The fossiliferous intervals are interpreted to be surface biofilm

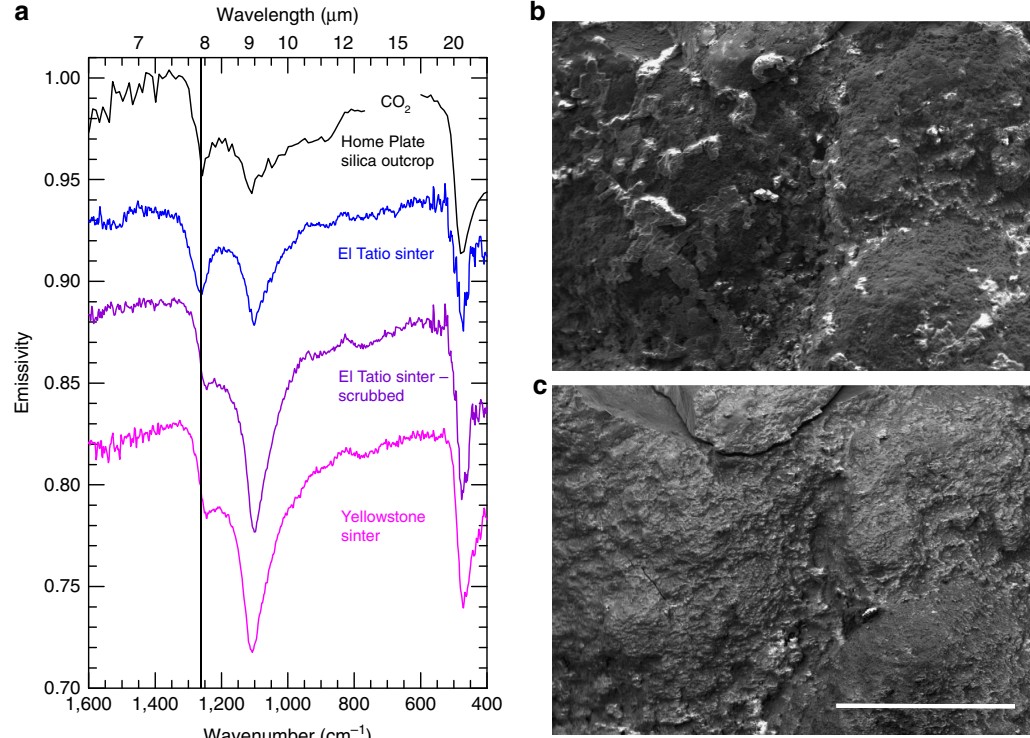

**Figure 5 | Spectral effect of halite on silica.** (**a**) Mini-TES spectrum of an opaline silica nodular outcrop adjacent to Home Plate (black, scaled by 2×; target Clara Zaph4, sol 1168, P3968) displays a strong feature at ~1,260 cm$^{-1}$ (vertical line) also found in halite encrusted silica sinter from El Tatio (sample ET3-3A) measured at 0° emission angle (blue; vertically offset). This feature is diminished substantially and slightly shifted after halite is removed (purple; vertically offset) and also in sinter that was never halite encrusted, like that from Yellowstone National Park (magenta; vertically offset). (**b**) SEM view of El Tatio sample with blue spectrum in **a** displays a patchy halite crust (lighter areas). (**c**) Same view as in **b** but with halite mostly removed by dissolution and scrubbing, yielding the purple spectrum in **a**. White scale bar represents 1 mm and applies to both **b**,**c**.

communities (Fig. 6b) that were entombed by opaline silica and incorporated into stromatolite profiles contributing to their accretion. In modern siliceous hot springs, such palisade microtextures have been reported widely from lower temperature, distal apron environments below ~35 °C (ref. 25), as well as ancient analogs from the Devonian of Australia[26]. Palisade microtextures also were documented previously among El Tatio silica oncoids and crusts[10,27]. Based on the suite of textural and microbial features apparent in thin sections and SEM images, we infer that El Tatio digitate silica structures are microbially mediated microstromatolites.

## Discussion

None of the observations by Spirit uniquely constrains the origin of the Home Plate nodular silica outcrops. However, the spectral evidence for encrustation by halite favours the role of chloride-bearing solutions rather than fumarolic gases. In some cases, fumaroles are known to produce sublimates of halite in minor amounts along with native sulfur and sulfur phases in greater abundance[28,29], but the latter have not been observed among the Home Plate silica outcrops[4].

Sodium-bearing alkali chloride waters are common in hot spring/geyser systems on Earth, with rarer examples of acid-sulfate-chloride waters also recognized[30,31]. The precipitation and accumulation of halite from such solutions requires evaporation to dryness and meteoric precipitation rates sufficiently low to avoid subsequent dissolution. These conditions are rare on Earth, but El Tatio is an example where halite is especially abundant among the silica sinters in discharge environments[32]. Conditions of high evaporation/low meteoric precipitation rates likely were present on Mars in the Late Noachian.

Based on analogy with El Tatio, the nodular and digitate silica structures, combined with evidence for halite crusts, substantially bolster the case for an origin of the Home Plate silica as sinter in a hot spring/geyser environment with precipitation from silica- and chloride-bearing waters. Evidence for silica sinter deposits on Mars is important given the known capacity of such rocks to capture and preserve microbes, making them ideal targets in the search for ancient life on Mars[33]. Furthermore, the fact that the silica phase at Home Plate has remained as opal-A rather than transforming to a mature polymorph like microquartz, attests to negligible diagenesis[4,34]. On Earth, opal-A is metastable. In New Zealand for example, the oldest recognized opal-A dominated sinter deposit dates to ~40,000 BP (ref. 35). In this context, we must consider the preservation history of the Home Plate opaline silica outcrops.

The Columbia Hills are inferred to be at least Late Noachian in age[36] and are embayed by flood basalts dated at 3.65 Ga based on crater size-frequency distribution[37]. Aeolian erosion has been the dominant geologic process since then[38]. One explanation for the preservation of the Home Plate silica outcrops over billions of years is that they were thinly buried for much of that time and only recently exposed by erosion. There is clear evidence for a succession of thin (decimeter scale) volcanic tephra deposits in the vicinity of Home Plate, including the eponymous feature[39], all capped by vesicular basalt boulders[3] (Fig. 1). Perhaps the silica outcrops formed relatively early in the volcanic succession, becoming buried first by tephra and then vesicular basalt, followed by exhumation via aeolian erosion. The lack of diagenesis of the silica is consistent with minimal burial and post-depositional conditions dominated by low water activity[34].

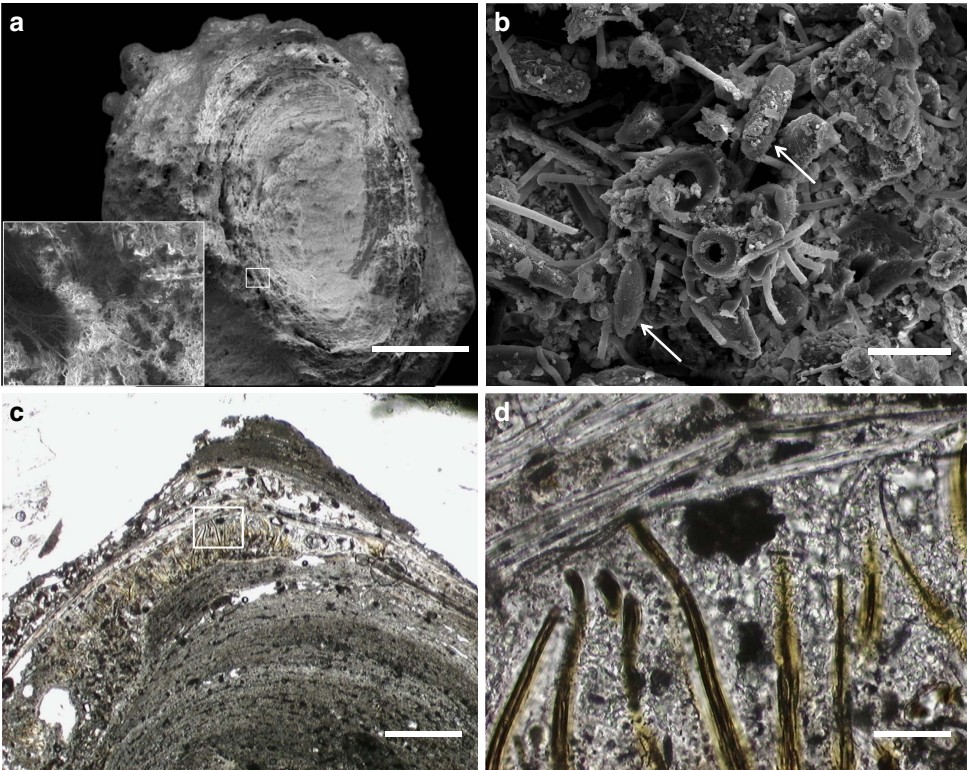

**Figure 6 | Microscopic views of El Tatio digitate silica structures.** (**a**) SEM image from a digitate structure broken off sample ET1-1A (Fig. 3f). Alternating non-porous and filamentous concentric laminae with fenestral porosity are evident. Inset highlights webs of silica-encrusted filaments within fenestral cavities. White scale bar represents 1 mm. (**b**) SEM image of a surface biofilm community showing silica-encrusted microbial filaments and sheaths, and spindle-shaped diatoms (arrows) occupying the outer surface of another digitate structure from the same sample. White scale bar represents 20 μm. (**c**) Photomicrograph of a transverse petrographic thin section through a digitate structure from a second El Tatio sample (ET1-1C). Microtextures include nonporous and fine-scale laminae, porous laminae with irregular to flattened fenestral cavities, and tufted palisade fabrics formed by silicified populations of filamentous cyanobacteria resembling *Calothrix* (family Rivulariaceae). White scale bar represents 500 μm. (**d**) Enlarged view from boxed area in **c** showing silicified *Calothrix* sheaths oriented roughly perpendicular to laminae. *Calothrix* sheaths (some containing cellular trichomes) have been heavily permeated by silica and are overlain by laminae containing silica encrusted fine filaments with orientations roughly parallel to laminae. White scale bar represents 50 μm.

On Earth, interbedded volcanic and hot spring/geyser deposits are common in the rock record and can include textures preserved following burial and exhumation[26,40,41]. Recent results from the Curiosity rover provide an example of buried sedimentary rocks on Mars having been exposed as recently as ~78 Myr ago (ref. 42). Such observations support a burial and exhumation hypothesis for the Home Plate silica outcrops.

The morphology of Home Plate digitate silica structures bears a strong resemblance to the microbially mediated microstromatolites at El Tatio. Siliceous microstromatolites are common features of hot spring/geyser systems on Earth[16,43,44], and the particular morphology of those at El Tatio might be due in part to halite accumulation. Although we have no microscale observations of the interiors of the Home Plate digitate silica structures, external shape generally is considered to be one of several distinguishing features of microbialites[45], including stromatolites[46]. Thus, a plausible hypothesis for the Home Plate digitate structures is that they are microstromatolites formed in hot spring/geyser discharge channels like those of El Tatio. However, determining the relative contribution of biotic and abiotic influences in the formation of a particular stromatolite can be quite difficult, and must be assessed on a case-by-case basis[16,47]. Entirely abiogenic internally laminated columnar structures have been synthesized via numerical modeling[46] and laboratory spray-paint deposition experiments[47], complicating the effort to interpret such structures in the Precambrian rock record on Earth. To fully test for biogenicity of the Home Plate digitate structures would require microscopic analyses like those we have applied to El Tatio samples.

Lacking information on internal microscale features of the Home Plate digitate silica structures, the hypothesis that they are microstromatolites arises from our interpretation of the integrated observations of geologic context, mineralogy and morphology down to mm-scale, all of which are consistent with a microbialite origin. Given the strong evidence that Home Plate nodular silica outcrops are sinter deposits, the abiotic production of digitate structures resembling those known to arise via biosedimentary processes among sinter deposits on Earth would require a fortuitous combination of constructional and/or erosional processes. Aeolian erosion might be a viable process to explain the digitate structures given the clear evidence for aeolian activity and erosion at Home Plate[38,48], but we have not encountered any natural or experimental examples of truly comparable structures produced by erosion. The preservation of a delicate surface texture like that on the Elizabeth Mahon outcrop (Figs 3e and 4e) seems to preclude the action of wind abrasion; presumably smoother surfaces would result from abrasion sufficient to sculpt digitate structures. Nevertheless, the absence of unequivocal evidence for life on Mars favours an abiotic origin by default. We note however, that none of the available observations actually precludes a biogenic origin for the Home Plate digitate silica structures, making them worthy of additional investigation.

The search for evidence of life on Mars remains a central focus of upcoming rover missions. In this context, NASA's Mars 2020 Science Definition Team defined a potential biosignature as 'an object, substance and/or pattern that might have a biological origin and thus compels investigators to gather more data before reaching a conclusion as to the presence or absence of life'[49]. Because we can neither prove nor disprove a biological origin for the microstromatolite-like digitate silica structures at Home Plate, they constitute a potential biosignature according to this definition.

A future rover mission could perhaps provide a more definitive assessment of biogenicity of Home Plate silica structures using instrumentation capable of identifying the presence or absence of microlaminations in exposed interiors. In combination with instrumentation capable of assessing the presence or absence of organic matter, the positive identification of internal microfabrics and detection of complex organic compounds would go a long way toward testing the hypothesis of biogenicity. However, because of the challenges in obtaining unambiguous evidence in situ, coordinated microscopic and compositional analyses of samples returned to laboratories on Earth may be required to reach a robust conclusion as to the presence or absence of past Martian life in these rocks.

## Methods

**Samples.** Opaline silica sinter samples used in this study were collected from hot spring and geyser discharge channels at El Tatio, Chile with permission from park officials. They were placed in plastic bags for transport without any fixatives or other preparation.

**SEM and EDS microscopy.** A scanning electron microscope (FEI XL30) integrated with energy dispersive X-ray spectroscopy was used to characterize sample micromorphology and semi-quantitative elemental chemistry. Both high vacuum and environmental SEM techniques were used in this study, with some samples sputter-coated with Au/Pd and others uncoated. They were attached to stubs using silver paint, copper tape, or carbon tape. Samples shown in Fig. 6a (uncoated) and 6b (coated) were run in high vacuum at 15 kV. The sample shown in Supplementary Fig. 5 was uncoated and run at 4.6 Torr at 25 kV.

**Thin section petrography.** Air-dried silica sinter samples were slabbed using a dry diamond saw, and transverse cut surfaces were cleaned with compressed air. Slabs were embedded in epoxy under a vacuum. Epoxy-embedded samples were sliced with a diamond oil-lubricated saw to produce doubly polished transverse one-inch round sections. Sections were ground in oil to a thickness of 30 microns and studied under a Nikon polarizing microscope. Images of key features were obtained under plane and crossed polarized light at magnifications ranging from 20× to 500×.

**Spectroscopy.** Laboratory thermal infrared emission spectra shown in Fig. 5a and Supplementary Figs 3 and 4 were measured with a Nicolet Nexus 670 spectrometer (modified for emission[50]) on natural surfaces of field samples except where noted as 'scrubbed' (described in main text). The samples were heated in an oven set to 80 °C for ~2 h before measurement and then placed on an apparatus designed to maintain this temperature during measurement. The 'field' spectrum shown in Supplementary Fig. 4 was obtained using a tripod mounted portable spectrometer (D&P µFTIR) oriented to measure the surface at an angle ~70° from the surface normal. The field of view included cobble-sized sinter fragments down to sand-sized particles.

**Data availability.** All Pancam approximate true colour images[51,52] are available at http://marswatch.astro.cornell.edu/pancam_instrument/true_color.html. All Microscopic Imager images are available at https://an.rsl.wustl.edu/mera/merxbrowser/. All other images and spectral data are available from the corresponding author upon request.

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

## Acknowledgements

This work was funded by NASA Mars Fundamental Research, Mars Data Analysis, and Exobiology programs. We thank the communities of Caspana and Toconce, Chile for allowing us access to field sites at El Tatio.

## Author contributions

Both S.W.R. and J.D.F. contributed to fieldwork at El Tatio. SEM work was led by S.W.R.; petrographic thin section work was led by J.D.F. Spectroscopy work was performed by S.W.R. The manuscript was written by S.W.R. with contributions from J.D.F.

## Additional information

**Competing financial interests:** The authors declare no competing financial interests.

