## [Peer Review File · Nature Communications]

Reviewer #1 (Remarks to the Author):

The authors have compared El Tatio sinters to potential Martian sinters. They provide convincing information that El Tatio is a good analogue for the possible sinters on Mars. They consider the high altitude factors (e.g., high evaporation rate, halite inclusion etc) and their influence of sinter formation. This is a very interesting and well written, original paper with excellent figures that will be of interest to many geoscientists.

The analytical methods used are all appropriate.

Is it possible to include a Thermal Infrared signature of a tephra unit (i.e., silica-rich) to show if there is any difference between the Thermal IR signatures of tephra and sinter? This would further confirm that the IR signatures are from sinters as suggested in the paper.

Regarding diagenesis - it would be good to include the location and age of the oldest known opal-A sinter deposit on Earth for comparison.

References are good and correctly cited except for two.

Lines 106 (ref. 8) and Line 198 (ref. 20) - all other references are shown by a superscript number.

The title is a little misleading. The paper provides strong evidence that the sinters at El Tatio are similar to those on Mars. Most of the analyses and results section is focussed on sinter samples from El Tatio that are very similar in appearance to those on Mars. I appreciate you do not have samples from Mars to do SEM/EDS and thin section work on but I think the paper title should be rewritten to include El Tatio. For example:

(1) Evidence strengthened for hot spring deposits on Mars with potential biosignatures based on comparison with El Tatio sinters, or (2) Striking resemblance of hot spring rocks from Mars with microbial-rich El Tatio sinters, or something similar.

Reviewer #2 (Remarks to the Author):

Ruff and Farmer make an interesting and convincing argument for the origin of certain nodular and digitate structures associated with the Homeplate outcrop in Gusev Crater as hydrothermal hot spring or geyserite deposits. In particular, they compare the morphology of the structures and their spectral signatures to deposits having similar morphology and spectral signatures from a Mars analogue environment, the El Tatio field in the Atacama Desert, Chile. These comparisons are detailed and convincing.

Since hydrothermal springs on Earth host microbes, even up to relatively high temperatures, the authors suggest that deposits of similar origin on Mars may do the same. Hydrothermal environments on Mars are a prime locality in the search for life, both for the ExoMars 2018 and the Mars 2020 missions. Thus, this study is highly relevant and timely.

Unfortunately, the authors have somewhat overstepped the boundary between reality and what they wish in their conclusion that, just because SOME biosignatures are preserved in recent hot spring/geyserite deposits on Earth, the features observed at Homeplate must necessarily be of biological origin.

The authors quote the review by Campbell et al (2015) and I quote verbatim (and extensively) certain passages to underline the fact that there is ALSO a strong abiotic component to hot spring/geyserite formation and that, although microbial biosignatures may be associated, this is not always the case and not necessarily preserved in deep geological time:

"The relative contribution of abiotic or biotic influence for a particular stromatolite must therefore be assessed on a case-by-case basis. This may be quite difficult to determine in environmental settings where biosignatures are cryptic or easily destroyed by taphonomic and diagenetic processes (see also Section 4.2)."

« White et al. (1964, p. B31) formally defined geyserite as microbanded opaline sinter of colloform, botryoidal or are somewhat delicate and can easily weather on dry, exposed sinter surfaces (Jones and Renaut, 2003), suggesting a possible taphonomic bias toward their destruction prior to incorporation into the geological record."

"Modern geyserite displays a characteristic feature - dense, even, very fine (500 nm-4 µm thick), alternating light/dark laminae - which variously has been interpreted as demarcating annual silica accumulation, daily precipitation, or individual eruptive cycles (Walter, 1972, 1976a; Lowe and Braunstein, 2003; Jones and Renaut, 2004)."

With respect to a potential microbial contribution to geyserite formation:

"While the geyseritic lithofacies may be a robust indicator of high temperature geothermal activity in terrestrial volcanic settings (Section 4.1), microbial fossil associations in the pre-Quaternary geyserite record are unknown.

"The specific role of microorganisms in terms of their volumetric contribution to the build-up of the solid sinter deposit around spring-vents and geyser mounds, and to what degree they influence micro/macro- textures of geyserite, are still under discussion. Most reports acknowledge the dominant role of abiogenic processes - rapid cooling driving evaporation and silica oversaturation - in the precipitation of noncrystalline opaline silica at geysers and vents (e.g., Göttlicher et al., 1998; Braunstein and Lowe, 2001; Lowe and Braunstein, 2003; McLoughlin et al., 2008). "

"Synthetic stromatolites mimicking biogenic stromatolites in siliceous sinter have been grown in the absence of microbes (McLoughlin et al., 2008). Near-vent splash zones have been considered to be sites of sinter accumulation as "a consequence of physico-chemical processes alone" (McLoughlin et al., 2008, p. 102).

"All these examples illustrate that some laminated, columnar, siliceous, Precambrian sedimentary structures mimic geyserite textures and potentially represent the silicifying build-ups of microbial biofilms that left subtle to no morphological signatures of their presence.

The take home message here is caution. In the first place, the features described from Homeplate could well have been formed by hot spring/geyserite processes. Secondly, it is possible that, if life were present in Gusev Crater at the time of formation of the deposits, it could have colonised the Homeplate hydrothermal environment. But the deposits would have been formed regardless of the presence of life. Thirdly, even if life was present, its traces may not have survived natural diagenetic processes. The authors show their preoccupation with the potential for biosignature preservation and go to pains to demonstrate the lack of significant diagenesis through comparison with the opal signatures (rather than microquartz) of the El Tatio deposit.

In conclusion, I think that the study by Ruff and Farmer is very valuable, worth while and timely. However, it suffers from a certain over-enthusiasm to push the potential for biogenicity associated with hot spring/geyserite deposits and, in overdoing so, it detracts from the excellent scientific and possible astrobiological potential of Homeplate as possible location for the search for life on Mars.

Reviewer #3 (Remarks to the Author):

Ruff and Farmer

«Evidence for hot spring deposits on Mars with potential biosignatures»

Recommendation: Reject

A. Summary of the key results

This manuscript compares nodular silica deposits from a modern hot spring deposit El Tatio in Chile to silica deposits from Gusev crater on Mars. The authors argue by analogy to the microlaminated stromatolitic structures at El Tatio that the Martian silica deposits could host potential microbial biosignatures. The martian data is not new, and the bulk of the data comes from the terrestrial site. The crux of the hypothesis is a qualitative morphological comparison between the two deposits, however the authors fail to explore, or acknowledge, the extent to which physio-chemical abiotic processes control stromatolite morphology. For the hypothesis suggested to be tenable, microtextural observations would be required from the Martian deposits. The case for a potential biosignature is weak. This data would be better presented and discussed in a disciplinary journal, I flag a number of questions and concerns below.

B. Originality and interest: if not novel, please give references

The interest is high, I am aware of only one other study of potential microbialite deposits on Mars (Russell et al. 1999), however the hypothesis is not substantiated in this study. The originality is not high, given that much of the Martian data has been presented previously, also that the El Tatio deposit has been previously characterized. The new aspect of this work is a photographic and spectral comparison between the two sites.

C. Data & methodology: validity of approach, quality of data, quality of presentation

Spectral and photographic data from Mars.

SEM-EDS imaging and photographic data from El Tatio, Chile.

I would like to see quantification of the size and shape of the Martian and the El Tatio nodular-digitate deposits - the current manuscript provides only a qualitative, photographic comparison. Also needs a description and measurement of the surface roughness of the Martian silica deposits, these appear to be quite porous and many readers would like to know if this relates to weathering, or mineralogy or what?

The new arguments for a NaCl coating on the Martian deposits to account for a previously unassigned peak in the mini-TES data needs to be checked by a spectral expert/reviewer, it is interesting but not in my opinion fundamental to the interpretation of the depositional environment. Groundwater sources need to be ruled out.

My main issue with the manuscript is that the macromorphologies of these deposits is only qualitatively documented and that we cannot see any evidence of lamination in the Martian deposits. Secondly, the argument that the macromorphology alone is suggestive of a stromatolitic and biological origin is flawed. Several studies have highlighted that the macromorphology of sinter deposits is controlled in large part by physio-chemical processes like flow rate. Also numerical and experimental studies of stromatolite growth have shown that diffusion limited processes control laminated macromorphology (e.g. Grotzinger and Knoll 1999, McLoughlin et al. 2008).

Microtextural observations give the strongest support for microbial participation in stromatolite construction - in the absence of such data from Mars this hypothesis is weak. I also disagree with the statement "solely abiotic processes have not yet been recognized forming the kind of terrestrial hot spring digitate silica structures shown herein". The aforementioned study "McLoughlin et al. 2008" raises the real possibility that micro-digitate stromatolites can form in sinter splash zones purely abiotically.

To be more nuanced, the paper needs to discuss and evaluate the probability that the deposits may host potential biosignatures against specific criteria (e.g. geological context, morphological evidence, and geochemical evidence) and discuss the types of approaches and instruments that

could obtain this data - would be suitable for a disciplinary journal like Astrobiology.

D. Appropriate use of statistics and treatment of uncertainties
Not relevant

E. Conclusions: robustness, validity, reliability

The potential biosignature conclusion is rather underwhelming and not wholly convincing: "The Home Plate silica structures can be interpreted plausibly as objects that might have a biological origin".

F. Suggested improvements: experiments, data for possible revision
Need to quantify the morphologies of the deposits being studied

EDS data, I assume this detector was on the SEM ? - there are more reliable techniques for confirming the presence of organic Carbon, e.g. Raman spectroscopy. Reliability of the light element abundance is uncertain, C in particular.

G. References: appropriate credit to previous work?
Yes

H. Clarity and context: lucidity of abstract/summary, appropriateness of abstract, introduction and conclusions
Clearly written on the whole.

Reviewer #4 (Remarks to the Author):

A. This manuscript presents results of new analysis of siliceous hot spring deposits from El Tatio, Chile, as an analogue for nodular opaline silica features discovered by the Spirit rover adjacent to 'Home Plate' on Mars.

B. This is a novel and highly interesting study, with important implications for future missions to search for life on Mars

C. The techniques and methods are perfectly valid.

D. N/A

E. Strong, exciting conclusions.

F. N/A

G. sufficient

H. Fairly well written. A number of minor corrections required for clarity and flow, as noted below.

First page (Title page): spelling of second author Farmer

Lines 2 and 3: Should the proper names be capitalised, ie Spirit Rover and Gusev Crater? e.g. Panama Canal. If so, make it a global change

Line 3: insert "deposited" after close parenthesis

Line 4: Insert "may have" before "formed", and "either" before "fumarole"

Line 6: Delete "outcrops", but insert "of the opaline silica" before "was".

Lines 6-7: Delete "We now have identified" and replace with "Here we report"

Line 8: insert "of an active hot spring" before "at El Tatio". Also, replace "the silica there" with "opaline silica at El Tatio"

Line 10: replace "found" with "show"

Line 11: insert comma after "El Tatio" and replace "that" with "which". Also, insert comma after Mars

Line 12: replace "(i.e., stromatolites)" with "in the form of microdigitate stromatolites"

Line 14: Clarify "a priori"...what prior? Do you mean a priori (italicized?). Not clear

Line 17: insert "both Earth and" before Mars and perhaps consider including a reference to support

the Earth preservation of biosignatures. A good review on this, for example, is Campbell et al., 2015 Earth Science Reviews.

Line 30, Figure 1 caption: insert "opaline silica" before "nodules"

Line 35, Figure 1 caption: Insert "outcrop, of the" before "portion"

Figure 1, part a: perhaps augment the photo with dashed lines showing the extent of silica nodule outcrop

Line 43: delete "salient"

Line 44: regarding stratiform relationship, does this mean overlying or underlying? Clarify

Line 45: Clarify what Halley subclass means. This is clearly a Mars team term, but not known to the general reader. Sounds like a type of submarine...

Lines 46-47: If the silica is a residual from the result of leaching, then there should be no primary textures of the original rock, as the leaching would have been so intense.

Lines 47-49: Explain this point: why? how?

Lines 50-51: Briefly explain what the status of this interpretation now - has it been abandoned? if so, why? If not, how do your new observations relate to this?

Line 53: insert "on the Spirit Rover" before "were used"

Lines 55-59: can you supply a Supplementary figure to show?

line 63: replace "So" with "Thus,"

Line 69: replace "unique in" with "rare, due to"

Line 82: insert "cored by" before "breccias" and delete "that consist" Line 91: replace "host" with "are coated by"

Line 92: delete "coatings". Also, by stating that the precipitation occurs as the grains are being transported along channels, are these oncoids?

Line 97: change substrates to substrate

Line 103: insert "nodular, digitate" before "structures"

Line 111: insert ", due to the limitation of" immediately after "structures"

Line 112: delete "was limited"

Line 120: rewrite to "deionised water, the action of which effectively removed..."

Line 121: replace "is evident from" with "was confirmed by"

Line 134, Fig 3 caption: insert "(light areas)" after "crust". Also, insert "mostly" before "removed"

Line 135: insert "El Tatio" before silica

Line 143: insert "El Tatio" before "ranges" (if this is right, or is it Mars?)

Line 144: insert "on Mars," before "akin"

Line 145: insert comma after sinter

Line 148: replace "has" with "was analysed for"

Line 193: insert "(Figs 4c, d)" after "laminae"

Line 208: delete period and capital T and replace with ", but t"

Line 215: insert "arising from low discharge and high evaporation rates" after "environments"

Line 216: do you mean low discharge instead of low meteoric precipitation?

Line 218; rewrite "Home Plate silica structures and" to "nodular silica structures combined with"

Line 219: insert comma after crusts. Also, insert "of the Home Plate silica" after "origin"

Line 220: replace "with" with ", precipitation from". Also, insert "on mars" at very end of line, after "deposits"

Line 221: insert "known" before "capacity"

Line 223: insert "on Mars" after "phase"

Line 233: replace "relatively early in" with "immediately prior to"

Line 234: replace "followed by exhumation" with "and then exhumed"

Line 238: replace "evident" with "preserved"

Line 246: delete ", given the suite of observations from Spirit,"

Line 248: insert "of Home Plate samples" before "like"

Line 249: replace "these" with "the Home Plate"

Line 252: delete "terrestrial hot spring"

Line 260: I suggest an addition to the end of the sentence "and are worthy of more detailed investigations via subsequent missions"

Fig S1 caption, 4th line: insert "on Mars," before "shown"

Fig S4 caption, 2nd line: insert "on Mars" after "Innocent Bystander"

Response to Reviewers' comments for Ruff and Farmer manuscript NCOMMS-16-08249-T

Reviewer #1 (Remarks to the Author):

The authors have compared El Tatio sinters to potential Martian sinters. They provide convincing information that El Tatio is a good analogue for the possible sinters on Mars. They consider the high altitude factors (e.g., high evaporation rate, halite inclusion etc) and their influence of sinter formation. This is a very interesting and well written, original paper with excellent figures that will be of interest to many geoscientists.

The analytical methods used are all appropriate.

Is it possible to include a Thermal Infrared signature of a tephra unit (i.e., silica-rich) to show if there is any difference between the Thermal IR signatures of tephra and sinter? This would further confirm that the IR signatures are from sinters as suggested in the paper.

Response: Although unaltered tephra and sinter are readily distinguishable, TIR spectroscopy alone cannot distinguish between opaline silica sinter and opaline silica residue from acidic alteration of tephra, basalt, rhyolite or any other silicate substrate. Thus the suggested comparison cannot be used to confirm that the Home Plate silica is unique to sinter, but we now include a new figure (Fig. 4) and text (lines 125-166) that present textural features found in silica sinter and Home Plate silica (beyond the morphological similarities already included). We also have added a new figure (Supplementary Fig. 1) that demonstrates that the Home Plate silica is spectrally matched with opal-A.

Regarding diagenesis - it would be good to include the location and age of the oldest known opal-A sinter deposit on Earth for comparison.

Response: We have added a sentence about the oldest opal-A recognized in a sinter deposit from New Zealand (~40,000 BP; lines 298-299)

References are good and correctly cited except for two.

Lines 106 (ref. 8) and Line 198 (ref. 20) - all other references are shown by a superscript number.

Response: These two citations follow the format used by Nature Communications where a superscript number is not used to avoid confusion in a particular context, like the degrees centigrade notation for the first example and inverse wavenumbers for the second.

The title is a little misleading. The paper provides strong evidence that the sinters at El Tatio are similar to those on Mars. Most of the analyses and results section is focussed on sinter samples from El Tatio that are very similar in appearance to those on Mars. I appreciate you do not have samples from Mars to do SEM/EDS and thin section work on but I think the paper title should be rewritten to include El Tatio. For example:

(1) Evidence strengthened for hot spring deposits on Mars with potential biosignatures based on comparison with El Tatio sinters, or (2) Striking resemblance of hot spring rocks from Mars with microbial-rich El Tatio sinters, or something similar.

Response: We have changed the title to accommodate the Reviewer's suggestion while maintaining the 15-word limit.

Reviewer #2 (Remarks to the Author):

Ruff and Farmer make an interesting and convincing argument for the origin of certain nodular and digitate structures associated with the Homeplate outcrop in Gusev Crater as hydrothermal hot spring or geyserite deposits. In particular, they compare the morphology of the structures and their spectral signatures to deposits having similar morphology and spectral signatures from a Mars analogue environment, the El Tatio field in the Atacama Desert, Chile. These comparisons are detailed and convincing.

Since hydrothermal springs on Earth host microbes, even up to relatively high temperatures, the authors suggest that deposits of similar origin on Mars may do the same. Hydrothermal environments on Mars are a prime locality in the search for life, both for the ExoMars 2018 and the Mars 2020 missions. Thus, this study is highly relevant and timely.

Unfortunately, the authors have somewhat overstepped the boundary between reality and what they wish in their conclusion that, just because SOME biosignatures are preserved in recent hot spring/geyserite deposits on Earth, the features observed at Homeplate must necessarily be of biological origin.

Response: Our intent is to present observations of features, both morphologic and spectral, that are similar between Home Plate and El Tatio silica occurrences. In so doing, we suggest the possible role of microbes in the case of the Home Plate silica structures given the microbial role in comparable El Tatio silica structures. We do not "wish" this to be true; rather we suggest it is plausible and satisfies an a priori definition for potential biosignatures.

That said, we have added emphasis to the possibility of abiotic processes in the formation of the silica structures at Home Plate along the lines of the quoted passages below. Please note the new title and see lines 12-13 in the Abstract and 317-349 in the substantially revised Discussion.

The authors quote the review by Campbell et al (2015) and I quote verbatim (and extensively) certain passages to underline the fact that there is ALSO a strong abiotic component to hot spring/geyserite formation and that, although microbial biosignatures may be associated, this is not always the case and not necessarily preserved in deep geological time:

"The relative contribution of abiotic or biotic influence for a particular stromatolite must therefore be assessed on a case-by-case basis. This may be quite difficult to determine in environmental settings where biosignatures are cryptic or easily destroyed by taphonomic and diagenetic processes (see also Section 4.2)."

« White et al. (1964, p. B31) formally defined geyselite as microbanded opaline sinter of colloform, botryoidal or are somewhat delicate and can easily weather on dry, exposed sinter surfaces (Jones and Renault, 2003), suggesting a possible taphonomic bias toward their destruction prior to incorporation into the geological record."

"Modern geyselite displays a characteristic feature - dense, even, very fine (500 nm-4 µm thick), alternating light/dark laminae - which variously has been interpreted as demarcating annual silica accumulation, daily precipitation, or individual eruptive cycles (Walter, 1972, 1976a; Lowe and Braunstein, 2003; Jones and Renault, 2004)."

With respect to a potential microbial contribution to geyselite formation:

"While the geyselite lithofacies may be a robust indicator of high temperature geothermal activity in terrestrial volcanic settings (Section 4.1), microbial fossil associations in the pre-Quaternary geyselite record are unknown.

"The specific role of microorganisms in terms of their volumetric contribution to the build-up of the solid sinter deposit around spring-vents and geyser mounds, and to what degree they influence micro/macro- textures of geyselite, are still under discussion. Most reports acknowledge the dominant role of abiogenic processes - rapid cooling driving evaporation and silica oversaturation - in the precipitation of noncrystalline opaline silica at geysers and vents (e.g., Göttlicher et al., 1998; Braunstein and Lowe, 2001; Lowe and Braunstein, 2003; McLoughlin et al., 2008). "

"Synthetic stromatolites mimicking biogenic stromatolites in siliceous sinter have been grown in the absence of microbes (McLoughlin et al., 2008). Near-vent splash zones have been considered to be sites of sinter accumulation as "a consequence of physico-chemical processes alone" (McLoughlin et al., 2008, p. 102).

"All these examples illustrate that some laminated, columnar, siliceous, Precambrian sedimentary structures mimic geyselite textures and potentially represent the silicifying build-ups of microbial biofilms that left subtle to no morphological signatures of their presence.

The take home message here is caution. In the first place, the features described from Homeplate could well have been formed by hot spring/geyselite processes. Secondly, it is possible that, if life were present in Gusev Crater at the time of formation of the deposits, it could have colonised the Homeplate hydrothermal environment. But the deposits would have been formed regardless of the presence of life. Thirdly, even if life was present, its

traces may not have survived natural diagenetic processes. The authors show their preoccupation with the potential for bioisgnature preservation and go to pains to demonstrate the lack of significant diagenesis through comparison with the opal signatures (rather than microquartz) of the El Tatio deposit.

Response: We agree with these points and have added text to emphasize the abiogenic potential (see previous response). Regarding the third point, addressing biosignature preservation potential is an obvious follow on to the observations we describe, especially in the context of exploring Mars for a record of life. The apparent lack of diagenesis evident in the persistence of the opal-A phase at Home Plate is noteworthy and perhaps a profoundly important consideration. It is not clear if the Reviewer is suggesting that we exclude discussion of this issue, and if so, why.

In conclusion, I think that the study by Ruff and Farmer is very valuable, worthwhile and timely. However, it suffers from a certain over-enthusiasm to push the potential for biogenicity associated with hot spring/geyserite deposits and, in overdoing so, it detracts from the excellent scientific and possible astrobiological potential of Homeplate as possible location for the search for life on Mars.

Response: We appreciate these comments. The manuscript is an attempt to draw attention to the newly recognized possible significance of a discovery made almost ten years ago, which apparently can be viewed as “over-enthusiasm”. With our revisions, we continue to show the striking similarities between observations from Home Plate and El Tatio and point out their potential significance, but we inject sufficient caveats to allow the reader to understand that abiotic processes might have been sufficient to produce all of features of the Home Plate silica deposits.

Reviewer #3 (Remarks to the Author):

Ruff and Farmer

«Evidence for hot spring deposits on Mars with potential biosignatures»

Recommendation: Reject

A. Summary of the key results

This manuscript compares nodular silica deposits from a modern hot spring deposit El Tatio in Chile to silica deposits from Gusev crater on Mars. The authors argue by analogy to the microlaminated stromatolitic structures at El Tatio that the Martian silica deposits could host potential microbial biosignatures. The martian data is not new, and the bulk of the data comes from the terrestrial site. The crux of the hypothesis is a qualitative morphological comparison between the two deposits, however the authors fail to explore, or acknowledge, the extent to which physio-chemical abiotic processes control stromatolite morphology. For the hypothesis suggested to be tenable, microtextural observations would be required from the Martian deposits. The case for a potential biosignature is weak. This data would be better presented and discussed in a disciplinary journal, I flag a number of questions and concerns below.

B. Originality and interest: if not novel, please give references

The interest is high, I am aware of only one other study of potential microbialite deposits on Mars (Russell et al. 1999), however the hypothesis is not substantiated in this study. The originality is not high, given that much of the Martian data has been presented previously, also that the El Tatio deposit has been previously characterized. The new aspect of this work is a photographic and spectral comparison between the two sites.

Response: No previous publication has presented observations from El Tatio to explain observations from Home Plate, nor has any previous work shown similarities between biomediated microstromatolites at El Tatio and silica structures on Mars. Thus our work is original.

C. Data & methodology: validity of approach, quality of data, quality of presentation

Spectral and photographic data from Mars.

SEM-EDS imaging and photographic data from El Tatio, Chile.

I would like to see quantification of the size and shape of the Martian and the El Tatio nodular-digitate deposits - the current manuscript provides only a qualitative, photographic comparison. Also needs a description and measurement of the surface roughness of the Martian silica deposits, these appear to be quite porous and many readers would like to know if this relates to weathering, or mineralogy or what?

Response: Our intent is to convey the qualitative similarity between El Tatio and Home Plate silica structures sufficient to support our hypothesis that they could have been formed by similar processes. This is now stated in lines 91-94. We believe that detailed quantification of size and shape of these structures at this point is unnecessary. The images are shown at common scales as described in captions and/or include scale bars, which should be sufficient to convey the qualitative similarities.

Regarding roughness/porosity of the Martian silica deposit, we have added a new, multi-panel figure and three paragraphs of text that explore the mm-scale textures evident in the highest resolution images in comparison to comparable textures of El Tatio sinter. Please see lines 125-166 and Fig. 4.

The new arguments for a NaCl coating on the Martian deposits to account for a previously unassigned peak in the mini-TES data needs to be checked by a spectral expert/reviewer, it is interesting but not in my opinion fundamental to the interpretation of the depositional environment. Groundwater sources need to be ruled out.

Response: The spectral evidence for NaCl coatings on opaline silica at Home Plate demonstrates consistency with an alkali-chloride hot spring/geyser setting like that at El Tatio. We don't claim that this observation rules out other settings, but suggest that it favors chloride-bearing solutions over fumarolic gases, the only other published candidate for the origin of the Home Plate silica. We are not aware, nor have we found examples in the literature, of groundwater-related processes that produce halite-

encrusted opaline silica deposits. We will include discussion of this possibility if an example is brought to our attention.

My main issue with the manuscript is that the macromorphologies of these deposits is only qualitatively documented and that we cannot see any evidence of lamination in the Martian deposits. Secondly, the argument that the macromorphology alone is suggestive of a stromatolitic and biological origin is flawed. Several studies have highlighted that the macromorphology of sinter deposits is controlled in large part by physio-chemical processes like flow rate. Also numerical and experimental studies of stromatolite growth have shown that diffusion limited processes control laminated macromorphology (e.g. Grotzinger and Knoll 1999, McLoughlin et al. 2008).

Response: We now explicitly note the lack of observations showing internal laminations among the Home Plate silica structures (lines 320-322 and 333-334), but also note that "external shape generally is considered to be one of several distinguishing features of microbialites, including stromatolites (Grotzinger and Knoll, 1999)" (lines 322-323). We do not claim that morphology alone is sufficient evidence to prove the structures are biomediated stromatolites. Instead, we frame our hypothesis as follows: "Lacking information on internal microscale features of the Home Plate digitate silica structures, the hypothesis that they are microstromatolites arises from our interpretation of the integrated observations of geologic context, mineralogy, and morphology down to mm-scale, all of which are consistent with a microbialite origin." Please see lines 317-349 in the Discussion to see how we've attempt to address the issue of biogenicity.

Microtextural observations give the strongest support for microbial participation in stromatolite construction - in the absence of such data from Mars this hypothesis is weak. I also disagree with the statement "solely abiotic processes have not yet been recognized forming the kind of terrestrial hot spring digitate silica structures shown herein". The aforementioned study "McLoughlin et al. 2008" raises the real possibility that micro-digitate stromatolites can form in sinter splash zones purely abiotically.

Response: We agree that microtextural observations are the only way to demonstrate biogenicity in any kind of stromatolitic structure. In the absence of such observations for the Martian structures, the strength of our hypothesis comes from the fact that their morphology, mineralogy, and volcanic setting allows for the possibility that they are biomediated microstromatolites by analogy with El Tatio. There is no available observation that can disprove this hypothesis. That said, we have removed the quoted statement noted above and replaced it with mention of the McLoughlin et al. work as evidence of abiotic construction of microstromatolites (lines 328-330).

To be more nuanced, the paper needs to discuss and evaluate the probability that the deposits may host potential biosignatures against specific criteria (e.g. geological context, morphological evidence, and geochemical evidence) and discuss the types of approaches and instruments that could obtain this data - would be suitable for a disciplinary journal like Astrobiology.

Response: We do in fact use geological context, morphological evidence, and geochemical evidence (mineralogy) to support our hypothesis of potential biosignatures. We agree that it would be helpful to have a discussion of approaches and instruments that could be used on a future mission to investigate the possible biogenicity of the Martian silica structures. Lines 357-366 have been added in response.

D. Appropriate use of statistics and treatment of uncertainties

Not relevant

E. Conclusions: robustness, validity, reliability

The potential biosignature conclusion is rather underwhelming and not wholly convincing: "The Home Plate silica structures can be interpreted plausibly as objects that might have a biological origin".

Response: We have simply applied an a priori definition of "potential biosignatures" to our observations. This is an intentionally conservative conclusion because we cannot, with existing data, prove the presence of definitive biosignatures. Likewise, there are no observations that disprove the possibility of biosignatures, so the suite of observations we present contributes to a valid claim of potential biosignatures, per definition. We have revised the wording to better convey this. Please see lines 350-356.

F. Suggested improvements: experiments, data for possible revision

Need to quantify the morphologies of the deposits being studied

Response: We think that such an effort could yield additional insights but is unnecessary to support our claim of similarity in scale and morphology between El Tatio and Home Plate silica structures. As indicated previously, we have now explicitly stated our rationale for qualitative comparisons (lines 91-94).

EDS data, I assume this detector was on the SEM ? - there are more reliable techniques for confirming the presence of organic Carbon, e.g. Raman spectroscopy. Reliability of the light element abundance is uncertain, C in particular.

Response: Yes, C was measured using EDS integrated with SEM, which was sufficient to confirm the presence of an organic coating indicated by field context (microbial mats) and hand sample discoloration. The semi-quantitative measurement is now described in the Methods section.

G. References: appropriate credit to previous work?

Yes

H. Clarity and context: lucidity of abstract/summary, appropriateness of abstract, introduction and conclusions

Clearly written on the whole.

Reviewer #4 (Remarks to the Author):

A. This manuscript presents results of new analysis of siliceous hot spring deposits from El Tatio, Chile, as an analogue for nodular opaline silica features discovered by the Spirit rover adjacent to 'Home Plate' on Mars.

B. This is a novel and highly interesting study, with important implications for future missions to search for life on Mars

C. The techniques and methods are perfectly valid.

D. N/A

E. Strong, exciting conclusions.

F. N/A

G. sufficient

H. Fairly well written. A number of minor corrections required for clarity and flow, as noted below.

First page (Title page): spelling of second author Farmer

Response: Nice catch!

Lines 2 and 3: Should the proper names be capitalised, ie Spirit Rover and Gusev Crater? e.g. Panama Canal. If so, make it a global change

Response: Spirit is one of the Mars Exploration Rovers, so "Spirit Rover" is not a proper name. According to the IAU, crater is implied in the proper name, hence not capitalized.

Line 3: insert "deposited" after close parenthesis

Response: We avoid this usage in this sentence to maintain a non-process description of the silica. "Deposit" implies a process (hot spring deposition) for which we build a case in subsequent descriptions of the Home Plate silica.

Line 4: Insert "may have" before "formed", and "either" before "fumarole"

Response: Changed to "...may have formed via either fumarole..."

Line 6: Delete "outcrops", but insert "of the opaline silica" before "was".

Response: Done.

Lines 6-7: Delete "We now have identified" and replace with "Here we report"

Response: Done.

Line 8: insert "of an active hot spring" before "at El Tatio". Also, replace "the silica there" with "opaline silica at El Tatio"

Response: New sentence – "Here we report remarkably similar features within active hot spring/geyser discharge channels at El Tatio in northern Chile, where halite (NaCl) crusts contribute infrared spectral characteristics that provide the best match yet to spectra from Spirit."

Line 10: replace "found" with "show"

Response: Done.

Line 11: insert comma after "El Tatio" and replace "that" with "which". Also, insert comma after Mars

Response: These suggested changes alter the intended meaning of the sentence. We have only examined the nodular and digitate structures that most closely resemble the Martian ones, so the suggested changes would allow for a more expansive interpretation than intended.

Line 12: replace "(i.e., stromatolites)" with " in the form of microdigitate stromatolites"

Response: The suggested change would exceed the 150-word abstract limit. In the interest of making the abstract more accessible to non-specialists and <150 words, we have removed the parenthetical phrase altogether.

Line 14: Clarify "a priori"...what prior? Do you mean a priori (italicized?). Not clear

Response: Changed to "a priori" (with italics).

Line 17: insert "both Earth and" before Mars and perhaps consider including a reference to support the Earth preservation of biosignatures. A good review on this, for example, is Campbell et al., 2015 Earth Science Reviews.

Response: Done. Used Walter, 1972 as the reference to support the statement that hot springs have "...long been targets in the search..."

Line 30, Figure 1 caption: insert "opaline silica" before "nodules"

Response: Done.

Line 35, Figure 1 caption: Insert "outcrop, of the" before "portion"

Response: This would be redundant given that the sentence starts out with "Nodular outcrop..."

Figure 1, part a: perhaps augment the photo with dashed lines showing the extent of silica nodule outcrop

Response: Done.

Line 43: delete "salient"

Response: Done.

Line 44: regarding stratiform relationship, does this mean overlying or underlying?

Clarify

Response: Changed to – "...overlying stratiform..."

Line 45: Clarify what Halley subclass means. This is clearly a Mars team term, but not known to the general reader. Sounds like a type of submarine...

Response: We have expanded on this with new text elaborating on the geologic context of the site. Please see lines 21-23.

Lines 46-47: If the silica is a residual from the result of leaching, then there should be no primary textures of the original rock, as the leaching would have been so intense.

Response: The text is quoted from a cited paper. Leaching often preserves textures, for example, acid-sulfate leached vesicular basalt from Sulphur Banks, HI still clearly displays vesicular textures even where interior crystal textures are destroyed.

Lines 47-49: Explain this point: why? how?

Response: Added – "...given that relatively Ti-rich silica sinters are known to occur on Earth⁵." The citation refers to Preston et al., 2008, Icarus.

Lines 50-51: Briefly explain what the status is of this interpretation now - has it been abandoned? if so, why? If not, how do your new observations relate to this?

Response: The sentence has been changed to – "An erosional origin for these structures has not been ruled out, but in light of our observations from El Tatio, a primary sedimentary origin needs to be considered." Additional discussion is included in Lines 340-346.

Line 53: insert "on the Spirit Rover" before "were used"

Response: Changed to – "Thermal infrared emission spectra of the Home Plate silica outcrops obtained by Spirit's Miniature Thermal Emission Spectrometer (Mini-TES; ~340 – 2000 cm⁻¹) were used to..."

Lines 55-59: can you supply a Supplementary figure to show?

Response: Yes, we have added Supplementary Fig. 1.

Line 63: replace "So" with "Thus,"

Response: Done.

Line 69: replace "unique in" with "rare, due to"

Response: Changed to – "The physical environment of El Tatio offers a rare combination of high elevation (~4300 m), low precipitation (<100 mm/yr), high mean annual evaporation rate (132 mm), common diurnal freeze-thaw⁸, and extremely high UV irradiance⁹."

Line 82: insert "cored by" before "breccias" and delete "that consist".

Response: Changed to: "Many El Tatio nodules are silica coated and cemented breccias composed of reworked pebbles of older, locally derived volcanic rocks and fragments of silica sinter." (Lines 94-96)

Line 91: replace "host" with "are coated by"

Response: Done

Line 92: delete "coatings". Also, by stating that the precipitation occurs as the grains are being transported along channels, are these oncoids?

Response: Deleted “coatings” here. In the case of coated grains within breccias, this does not fit the description of oncoids as described at El Tatio by Jones and Renaut, 1997.

Line 97: change substrates to substrate

Response: Done

Line 103: insert "nodular, digitate" before "structures"

Response: Done

Line 111: insert ", due to the limitation of" immediately after "structures"

Response: Changed to – “Unfortunately, the resolution of Spirit’s microscopic imaging capability precludes our ability to observe any microscale features among the Home Plate silica structures.” (Lines 174-176)

Line 112: delete "was limited"

Response: See previous

Line 120: rewrite to "deionised water, the action of which effectively removed..."

Response: Unchanged. This change would produce an unnecessarily long sentence (4 lines) without improving readability or adding clarification.

Line 121: replace "is evident from" with "was confirmed by"

Response: Done

Line 134, Fig 3 caption: insert "(light areas)" after "crust". Also, insert "mostly" before "removed"

Response: Done

Line 135: insert "El Tatio" before silica

Response: This paragraph describes observations by the Mini-TES instrument of Home Plate silica. We have clarified this. (Line 203)

Line 143: insert "El Tatio" before "ranges" (if this is right, or is it Mars?)

Response: See above.

Line 144: insert "on Mars," before "akin"

Response: Clarified by adding “Home Plate” before “silica outcrops” in the previous sentence.

Line 145: insert comma after sinter

Response: Unchanged. The sentence is sufficiently short and direct that a comma seems unnecessary.

Line 148: replace "has" with "was analysed for"

Response: Changed to – “Unfortunately, none of the outcrop targets displaying a strong $\sim 1260\text{ cm}^{-1}$ feature was measured by the APXS, precluding a direct comparison between the two instruments.” (Lines 215-217)

Line 193: insert "(Figs 4c, d)" after "laminae"

Response: Done (now Figs 6c,d).

Line 208: delete period and capital T and replace with ", but t"

Response: Done

Line 215: insert "arising from low discharge and high evaporation rates" after "environments"

Response: Did not change (see next response).

Line 216: do you mean low discharge instead of low meteoric precipitation?

Response: We mean low meteoric precipitation. The reasons for the presence of halite coatings on sinter at El Tatio include the abundance of dissolved chloride salts in the waters, the high evaporation rates, and the low meteoric precipitation that could re-dissolve the precipitated halite. A relationship to low discharge rates has not been documented.

Line 218; rewrite "Home Plate silica structures and" to "nodular silica structures combined with"

Response: Done

Line 219: insert comma after crusts. Also, insert "of the Home Plate silica" after "origin"

Response: Done

Line 220: replace "with" with ", precipitation from". Also, insert "on mars" at very end of line, after "deposits"

Response: Done

Line 221: insert "known" before "capacity"

Response: Done

Line 223: insert "on Mars" after "phase"

Response: Changed to “at Home Plate”.

Line 233: replace "relatively early in" with "immediately prior to"

Response: Unchanged; it's not clear whether the altered ash(?) deposit (Halley Subclass) on which the silica occurs is part of the volcanic succession.

Line 234: replace "followed by exhumation" with "and then exhumed"

Response: Unchanged to avoid duplicating the previous use of “and then”.

Line 238: replace "evident" with "preserved"

Response: Done

Line 246: delete ", given the suite of observations from Spirit,"

Response: Done

Line 248: insert "of Home Plate samples" before "like"

Response: Done

Line 249: replace "these" with "the Home Plate"

Response: Done

Line 252: delete "terrestrial hotspring"

Response: Done

Line 260: I suggest an addition to the end of the sentence "and are worthy of more detailed investigations via subsequent missions"

Response: In the heavily revised Discussion section, we have added the sentence (lines 347-349) – “We note however, that none of the available observations actually preclude a biogenic origin for the Home Plate digitate silica structures, making them worthy of additional investigations.” Discussion of future missions and measurements is now included in the last paragraph.

Fig S1 caption, 4th line: insert "on Mars," before "shown"

Response: This figure has now been incorporated into Fig. 3; the edit has been made there.

Fig S4 caption, 2nd line: insert "on Mars" after "Innocent Bystander"

Response: This figure has now been incorporated into Fig. 4 and the caption has been revised.

Reviewer #1 (Remarks to the Author):

Points by reviewer 1 are adequately addressed.

This is a well written paper that will be of interest to many different scientific disciplines.

Reviewer #2 (Remarks to the Author):

Rereview of Ruff and Farmer

Ruff and Farmer have taken great pains to address all the concerns of the all reviewers and to nuance their biogenic conclusions regarding the Home Plate silica deposits. This manuscript is a pleasure to read.

As it stands, the manuscript is acceptable for publication.

Just a note to the authors that they might find interesting, Foucher and Westall (Foucher, F. and Westall, F. 2013. Raman imaging of metastable opal in carbonaceous microfossils of the 700-800 Ma old Draken Formation, *Astrobiology*, 13, 57-67) noted the stabilisation of metastable opal up to 1.9 ga through association with biogenic kerogen.

Reviewer #3 (Remarks to the Author):

This manuscript has been carefully revised. The new title, abstract and discussion reflected a more careful assessment of the potential biogenicity of these structures and the type of data that would be required to confirm the biogenicity of these opaline silica deposits on Mars beyond doubt. Is a stimulating hypothesis, deserved publication now it has been revised.

Response to Reviewers' comments for Ruff and Farmer manuscript NCOMMS-16-08249B

Reviewer #1 (Remarks to the Author):

Points by reviewer 1 are adequately addressed.
This is a well written paper that will be of interest to many different scientific disciplines.

Response: Thank you.

Reviewer #2 (Remarks to the Author):

Re-review of Ruff and Farmer

Ruff and Farmer have taken great pains to address all the concerns of the all reviewers and to nuance their biogenic conclusions regarding the Home Plate silica deposits. This manuscript is a pleasure to read.

As it stands, the manuscript is acceptable for publication.

Just a note to the authors that they might find interesting, Foucher and Westall (Foucher, F. and Westall, F. 2013. Raman imaging of metastable opal in carbonaceous microfossils of the 700-800 Ma old Draken Formation, *Astrobiology*, 13, 57-67) noted the stabilisation of metastable opal up to 1.9 ga through association with biogenic kerogen.

Response: We appreciate the note and are aware of this work. We have opted to not include any mention of this work in our manuscript to avoid any additional provocative subject matter given that our work already is provocative. Also, the work of Foucher and Westall applies to microscopic opal and its preservation, which is not the case for the Home Plate opaline silica.

Reviewer #3 (Remarks to the Author):

This manuscript has been carefully revised. The new title, abstract and discussion reflected a more careful assessment of the potential biogenicity of these structures and the type of data that would be required to confirm the biogenicity of these opaline silica deposits on Mars beyond doubt. Is a stimulating hypothesis, deserved publication now it has been revised.

Response: Thank you.